# Lipid Metabolism Disorder in Cerebrospinal Fluid Related to Parkinson’s Disease

**DOI:** 10.3390/brainsci13081166

**Published:** 2023-08-04

**Authors:** Jiewen Qiu, Lijian Wei, Yilin Su, Yuting Tang, Guoyou Peng, Yimin Wu, Yan He, Hanqun Liu, Wenyuan Guo, Zhuohu Wu, Pingyi Xu, Mingshu Mo

**Affiliations:** 1Department of Neurology, The First Affiliated Hospital of Guangzhou Medical University, Guangzhou 510120, China; 2Department of General Medicine, Fengxian Community Health Service Center, Shanghai 210499, China

**Keywords:** Parkinson’s disease, biomarker, cerebrospinal fluid, lipid metabolism

## Abstract

Background: Abnormal accumulation of lipids is found in dopamine neurons and resident microglia in the substantia nigra of patients with Parkinson’s disease (PD). The accumulation of lipids is an important risk factor for PD. Previous studies have mainly focussed on lipid metabolism in peripheral blood, but little attention has been given to cerebrospinal fluid (CSF). We drew the lipidomic signature in CSF from PD patients and evaluated the role of lipids in CSF as biomarkers for PD diagnosis. Methods: Based on lipidomic approaches, we investigated and compared lipid metabolism in CSF from PD patients and healthy controls without dyslipidaemia in peripheral blood and explored the relationship of lipids between CSF and serum by Pearson correlation analysis. Results: A total of 231 lipid species were detected and classified into 13 families in the CSF. The lipid families, including phosphatidylcholine (PC), sphingomyelin (SM) and cholesterol ester (CE), had significantly increased expression compared with the control. Hierarchical clustering was performed to distinguish PD patients based on the significantly changed expression of 34 lipid species. Unsupervised and supervised methods were used to refine this classification. A total of 12 lipid species, including 3-hydroxy-dodecanoyl-carnitine, Cer(d18:1/24:1), CE(20:4), CE(22:6), PC(14:0/18:2), PC(O-18:3/20:2), PC(O-20:2/24:3), SM(d18:0/16:0), SM(d18:2/14:0), SM(d18:2/24:1), SM(d18:1/20:1) and SM(d18:1/12:0), were selected to draw the lipidomic signature of PD. Correlation analysis was performed and showed that the CE family and CE (22:6) in CSF had a positive association with total cholesterol in the peripheral blood from PD patients but not from healthy controls. Conclusions: Our results revealed that the lipidomic signature in CSF may be considered a potential biomarker for PD diagnosis, and increased CE, PC and SM in CSF may reveal pathological changes in PD patients, such as blood–brain barrier leakage.

## 1. Background

The major pathological hallmarks of Parkinson’s disease (PD) include degeneration and apoptosis of dopaminergic (DA) neurons, abnormal aggregation of α-synuclein (α-syn) and microglial activation in the substantia nigra (SN) [1]. Recent studies have shown that lipid and lipoprotein metabolism are involved in neurodegenerative disorders such as PD [2]. The concentration of glycosphingolipid substrates in the brain is abnormally elevated with ageing in familial and sporadic PD patients [3]. In the late stage of PD, lipid vesicles are found in Lewy bodies (LBs), accompanied by the deposition of α-syn, membrane fragments and cytoskeleton elements in the brain [2,4]. Regarding genetic factors, some mutations in the glucosylceramidase-beta (GBA) gene, encoding the lysosomal hydrolase glucocerebrosidase (GCase), are high risk factors for PD and dementia with LBs (DLBs) [5,6].

Lipid metabolism plays an essential role in the occurrence and development of brain diseases [7]. Deficiency in the biosynthesis and metabolic degradation of sphingolipids is involved in Gaucher disease, hereditary sensory neuropathy type 1 and infantile epilepsy syndrome [8,9,10]. Lipid metabolism disorders are also found in mental diseases, including bipolar disorder and schizophrenia [11,12]. In the early stage of Alzheimer’s disease (AD), cerebroside and sulfatide are dramatically reduced in the grey matter, and ceramide is elevated in white matter before clinical symptoms [13]. A clinical study of 29 samples from PD patients with a GBA mutation found that monohexylceramide, ceramide and sphingomyelin were increased and that phosphatidylic acid, plasminogen phosphatidylethanolamine, phosphatidylethanolamine and acyl phosphatidylglycerol were decreased in serum [14]. Under normal conditions, α-syn protein has amphipathic N-terminal amino acid domains to bind with special lipids [15]. Some mutations in the α-syn gene, including A30P, E46K, G51D, H50Q and A53T, which drive familial PD, are located in the lipid binding domain [16]. These mutations induce the inability of α-syn to bind with lipids, and the abundant lipids are deposited in LBs [16,17]. The impaired protein–lipid interaction may affect α-syn oligomerisation and aggregation, the activity of cytoplasmic and lysosomal enzymes, lipid metabolism and vesicular transport and ultimately lead to the degeneration of DA neurons [18].

Even now, the relationship between lipid metabolism in the central nervous system (CNS) and PD remains the subject of controversy. Research on lipid metabolism in the cerebrospinal fluid (CSF) is rare, especially in PD. Compared with peripheral blood, the lipid signature in CSF has better accuracy and sensitivity for estimating pathological changes in the CNS [19]. Thus, this study aims to explore the lipid metabolism in CSF from PD patients, analyse its role in disease and evaluate its association between serum and CSF lipids to find a helpful tool for PD diagnosis.

## 2. Methods

### 2.1. Study Population

A total of 152 Han Chinese subjects, including 101 sporadic PD patients and 51 healthy controls matched by age and sex, were recruited from the First Affiliated Hospital of Guangzhou Medical University from January 2018 to August 2021 as a cross-sectional study. The diagnosis was made by 2 neurologists following the Movement Disorder Society Clinical Diagnostic Criteria for PD [20]. The included patients had not accepted anti-PD treatments or antihyperlipidemic medicines before. The exclusion criteria included a medical history of aphasia, consciousness disorder, mental disorder, impaired brain function, dementia and metabolic disease such as hyperlipidaemia. The PD symptoms were evaluated using the Unified Parkinson’s Disease Rating Scale (UPRDS) and Hoehn and Yahr (H&Y) stage. The Montreal Cognitive Assessment test (MoCA) and Mini-Mental State Examination (MMSE) were performed to assess the global cognitive functions of all participants. The study protocols were approved by the local ethics committee at the First Affiliated Hospital of Guangzhou Medical University. All samples were analysed after obtaining informed consent from the participants. A total of 27 CSF samples without GBA mutations were obtained using lumbar puncture; overall, 152 serum samples were obtained, and 1 CSF sample of PD was excluded due to the development of hypercholesterolemia (Figure 1).

### 2.2. Peripheral Blood and CSF Collection

Peripheral blood samples were collected from all participants within 2 weeks of CSF collection. After blood samples were drawn into tubes containing an anticoagulant of citric acid, the plasma was obtained by centrifugation at 500× *g* for 10 min at 4 °C. The total cholesterol (TC), triglycerides (TGs), low-density lipoprotein cholesterol (LDL-C) and high-density lipoprotein cholesterol (HDL-C) were determined within the first 24 h of admission into the hospital using a ChemWell chemistry analyser (Awareness Tech, Palm City, FL, USA). The CSF sample was centrifuged at 3000 rpm at 4 °C for 5 min. A 50 μL sample was used and homogenised with methanol, methyl tert-butyl ether and internal standard mixture. A total of 500 μL of water was added and mixed, then 500 μL supernatant was extracted and concentrated. The powder was dissolved with 100 μL mobile phase B, and stored at −80 °C. The dissolving solution was mixed with the CSF sample for LC-MS/MS analysis. The maximum number of freeze–thaw cycles for detection was 2.

### 2.3. UPLC–MS/MS Analysis

The method used for lipid extraction from CSF has been reported before [19]. After thawing, mixing and centrifuging (3000 r/min) at 4 °C for 5 min, 50 μL of CSF sample was mixed with 1 mL solution containing Bligh–Dyer extraction and lipid standards for internal standardisation (Appendix A). After mixing, the samples were ultrasonicated in a water bath with a frequency of 40 kHz and a power of 100 W for 5 min. Then, 500 μL Milli-Q water was added, mixed and centrifuged (12,000 r/min) at 4 °C for 10 min. Finally, 500 μL supernatant was collected and mixed with 100 μL mobile phase B. Mobile phases A (60/40 acetonitrile/water) and B (10/90 acetonitrile/water) both contained 0.04% acetic acid and 10 mmol/L ammonium formate. The quality control was a pool with 20 µL of each sample. Ultra-performance liquid chromatography (UPLC) using a Shim-pack UFLC SHIMADZU CBM30A (Shimadzu, Kyoto, Japan) was coupled to the tandem mass spectrometry (MS/MS) using a QTRAP5500 (SCIEX, Framingham, MA, USA) as the detection system. The lipid extracts were analysed using UPLC–MS as reported previously [19].

The chromatographic separation was performed on an ACQUITY BEH C18 chromatography column (Waters Corporation; 2.1 mm × 100 mm, 1.8 μm). The column temperature was maintained at 40 °C, and mobile Phases A and B both contained 0.04% acetic acid. The injection volume was 5.0 μL, and the flow rate was 0.4 mL/min. The gradient elution program was set as follows: 10% B at the beginning; 10–33% B between 0 and 7 min, 33–56% B between 7 and 14 min, 56–100% B between 14 and 21 min; then, 100% B for 23.5 min; and 1.5 min for one recycle time. The desolvation temperature at electrospray ionisation was 550 °C. The MS voltage was set to 5500 V in positive mode and −4500 V in negative mode. The pressures of the curtain gas, gas II and ion source gas were set at 25 psi, 60 psi and 55 psi, respectively [19,21]. The ion pairs were detected using optimised collision energy and declustering potential in the triple quadrupole UPLC–MS/MS (Appendix A). The whole detection of lipids was finished in two batches.

### 2.4. Lipidomic Analysis

A widely targeted metabolome method based on the Metware database (Metware Biotechnology Co., Ltd., Wuhan, Hubei, China) was performed in UPLC–MS/MS analysis [22]. The ion pair information, retention time (RT) and secondary spectrum data of the detected substances were collected for qualitative analysis. The multiple reaction monitoring mode (MRM) of triple quadrupole mass spectrometry was performed for semi-quantitative analysis. MRM data analysis was performed using Analyst 1.6.3 software (AB SCIEX, Framingham, MA, USA) and MultiQuant (version 3.0, AB SCIEX). For peak area determinations, the data of all metabolites were extracted, and the individual area of the same metabolite was normalised to the integrated area of all peaks. The quality control (QC) was mixed with all sample extracts. One QC sample was inserted into every ten samples. The repeatability of metabolite extraction and detection was evaluated by the total ion current (TIC) of various QC samples. Regarding lipid families, the lipid species were classified as eicosanoids, free fatty acid (FFA), phosphatidylserine (PS), 3-hydroxy-dodecanoyl-carnitine (CAR), cholesterol ester (CE), ceramide (CER), diglyceride (DG), lysophosphatidylcholine (LPC), monoglyceride (MG), phosphatidylcholine (PC), phosphatidylethanolamine (PE), sphingomyelin (SM) and triglyceride (TG). Each concentration of lipid species was added together as the concentration of lipid families for analysis [22,23].

## 3. Statistical Analysis

The clinical parameters of participants underwent statistical analysis using the chi-square test, unpaired *t* test and Mann–Whitney test after the normality test using SPSS 16.0 (IBM, Armonk, NY, USA). The data with a normal distribution are expressed as the mean ± standard deviation. The data without a normal distribution are expressed as the median ± interquartile range. Statistical significance was considered when *p* < 0.05. The strength of linear relationships between PD and healthy controls was analysed with Pearson correlations using GraphPad Prism (GraphPad Prism^®^ Software version 8.0.2 for Windows; La Jolla, CA, USA). The significance of each variable in the Pearson correlation and linear mixed-effects model was assessed with the level at 0.05. In lipidomic analysis, principal component analysis (PCA), partial least squares-discriminant analysis (PLS-DA) and orthogonal partial least-squares discrimination analysis (OPLS-DA) were performed using SIMCA version 15 software (Umetrics, Umeå, Sweden) and MetaboAnalyst software (Version 5.0, https://www.metaboanalyst.ca/, accessed on 11 February 2023). The distribution of metabolites are shown as a volcano plot created by GraphPad Prism (GraphPad Prism^®^ Software version 8.0.2 for Windows; La Jolla, CA, USA) and Origin software (Version 2022, OriginLab Inc., Northampton, MA, USA). The metabolites with significant differences between the two groups were selected by −log_10_ (false-discovery-rate) and absolute log2 fold-change and used for cluster analysis in Origin software (Version 2022, OriginLab Inc., Northampton, MA, USA). The individual expression of lipids is shown as violin plots created by Hiplot (https://hiplot.com.cn, accessed on 11 February 2023).

## 4. Results

### 4.1. Lipidomic Signature of Serum in PD

The demographic characterisation of the PD cases and controls is presented in Table 1. A total of 101 sporadic PD patients and 51 age- and sex-matched healthy controls were recruited as two groups. The PD group recruited patients diagnosed at an early H&Y stage of disease with a short disease duration (3.21 ± 1.98 years) and a moderate MDS-UPDRS-III score (64.09 ± 26.53). Compared with the controls, the PD group had significantly lower MoCA scores (unpaired *t* test, *p* = 0.006) but no difference in MMSE scores (*p* = 0.147). Regarding the lipidomic signature of serum, the PD patients had lower concentrations of TG (*p* = 0.663), TC (*p* = 0.903) and LDL (*p* = 0.666) and higher concentrations of HDL (*p* = 0.512) in serum without significance.

### 4.2. Lipid Composition from CSF in PD

In the above population of donated serum, a total of 17 sporadic PD patients and 10 controls provided CSF for lipid analysis. The PD group had no difference in sex (*p* = 0.350), age (*p* = 0.226), education (*p* = 0.557), MMSE score (*p* = 0.322) or MOCA score (*p* = 0.089), or the concentration of TG (*p* = 0.538), TC (*p* = 0.963), LDL (*p* = 0.767) and HDL (*p* = 0.404) in serum compared with the control (Table 2). The global lipidomic differences in CSF were detected using a nontargeted approach. This method included 231 lipid species presented in two groups. These 231 lipid species were classified into 13 families, including 2 eicosanoids, 18 FFAs, 1 PS, 8 CARs, 4 CEs, 8 CERs, 17 DGs, 12 LPCs, 2 MGs, 39 PCs, 8 PEs, 19 SMs and 93 TGs. According to the rate of intension, the TG, FFA, LPC and PC families accounted for 36%, 24%, 19% and 13% in the control group and 28%, 21%, 16% and 25% in the PD group, respectively (Figure 2A,B). Hierarchical clustering analysis with all the lipid families was performed to draw a complete overview (Figure 2C). More details are presented by box plot with changed expression (Figure 2D). Compared with the control, the upregulated fold changes (FCs) in the expression of the PC, SM and CE families were 2.20, 3.17 and 4.26 in PD. The PC, SM and CE families showed significantly increased expression (*p* = 0.0006, 0.0001, 0.0002, respectively), but the other lipid families did not (Figure 3, Appendix A).

### 4.3. Lipidomic Profile in CSF from PD Patients

To define the lipidomic signature from CSF in PD patients, hierarchical clustering was performed using 231 lipid species to obtain a complete overview (Figure 4A, Appendix A). According to the selection criteria, a total of 34 lipid species met the criteria of adjusted *p* < 0.01 and log_2_FC > 1.5 (Figure 4B). These lipid species were included to draw the characteristeristic lipidomic signature in CSF from PD patients (Figure 4C). Among them, 12 lipid species met the log_2_FC > 2, and their expression details are presented by box plot (Figure 4D). Compared with the control, the significantly increased expression in lipid species included 3-hydroxy-dodecanoyl-carnitine with *p* = 2.12 × 10^−6^, Cer(d18:1/24:1) with *p* = 3.06 × 10^−6^, CE(20:4) with *p* = 1.04 × 10^−4^, CE(22:6) with *p* = 4.31 × 10^−5^, PC(14:0/18:2) with *p* = 1.10 × 10^−5^, PC(O-18:3/20:2) with *p* = 2.60 × 10^−6^, PC(O-20:2/24:3) with *p* = 4.39 × 10^−5^, SM(d18:0/16:0) with *p* = 8.24 × 10^−5^, SM(d18:2/14:0) with *p* = 2.06 × 10^−6^, SM(d18:2/24:1) with *p* = 4.37 × 10^−5^, SM(d18:1/20:1) with *p* = 2.12 × 10^−5^ and SM(d18:1/12:0) with *p* = 2.95 × 10^−6^ (Figure 5).

### 4.4. Different Pattern Recognition for the Lipidomic Signature in CSF

To evaluate whether a specific lipidomic signature in CSF from PD could be determined, data were analysed using unsupervised and supervised methods. PCA, as an unsupervised method, showed that PC-1 and PC-2 accounted for 48.0% and 20.2% of the total variation, respectively (Figure 6A). In supervised methods, such as PLS-DA, 20.1% of the total variance by component-1 and 19.7% of the total variance by component-2 can be explained (Figure 6B). It showed a valuable clusterisation of PD and healthy subjects with a specific lipidomic signature. This lipidome analysis could accurately diagnose PD in 96.3% of the cases, and its maximum value of Q^2^ of 0.74 can be obtained by using four components with an R^2^ of 0.92 (Figure 6E). The OPLS-DA method was also performed to separate the PD patients from healthy controls (Figure 6C), and its R^2^X, R^2^Y and Q^2^ were 0.190, 0.786 and 0.649, respectively. These results indicated that 19.0% of the X variables could be used to describe 78.6% of the variation between the two groups, and this model had 64.9% of the average predictability. The permutation test was used, and its Q^2^ intercept value was −0.878, which suggested that the quality and robustness of the OPLS-DA model was statistically effective (Figure 6D). The lipids with the top 12 variable importance in projection (VIP) values were identified and are shown in the VIP score plot (Figure 6F).

### 4.5. Relationship of Cholesterol Ester and Total Cholesterol in CSF from PD Patients

CE is a key form of TC in peripheral blood. The concentration of CE in CSF and brain is blocked by the blood–brain barrier (BBB) and is not affected by TC in peripheral blood [24]. CE showed significantly upregulated expression in CSF from PD patients in the above results. Here, the relationships of CE and TC in peripheral blood and CSF were analysed. Pearson correlation analysis was performed, and the results showed that the expression of CE in CSF had no correlation with the concentration of TC in peripheral blood from healthy controls; the Pearson correlation value was −0.193, with a *p* value of 0.593. In PD patients, the positive correlation between CE in CSF and TC in peripheral blood was significant, with *p* value of 0.005 and r value of 0.416 (Figure 7A). Regarding lipid species, the expression of CE (22:6) in CSF had a positive correlation with TC in peripheral blood, with a *p* value of 0.011 and r value of 0.356, but not CE (20:4), with a *p* value of 0.161 in PD. In healthy controls, the expression of CE (22:6) and CE (20:4) both had no significant correlation with TC in peripheral blood, with *p* values of 0.614 and 0.694, respectively (Figure 7B,C). This different association of CE between PD and healthy controls possibly suggested that the impairment of the BBB may occur in PD.

## 5. Discussion

PD exhibits diverse clinical presentations and is clinically indistinguishable in the early stages, which increases the difficulty of early diagnosis [25]. Biomarkers may help with PD diagnosis, but few ideal candidates has been found in serum or CSF for early diagnosis [26]. Due to the BBB, CSF biomarkers have better accuracy and sensitivity for estimating pathological changes in the CNS than serum biomarkers [24]. Lipidomics in CSF may be a powerful tool to evaluate brain lesions in PD [26]. In our study, we used lipidomics in CSF to demonstrate that PD patients present a special lipid signature and found that increased expression of the CE, PC and SM families and species in CSF may be suitable biomarkers for the evaluation of brain dysfunction, such as leaky BBB, in PD.

The lipids in CSF detected in this study are classified into 13 lipid families. Among them, PC, SM and CE had significantly upregulated expression compared with the control. PC is one of the most abundant phospholipids in most eukaryotic membranes, including the mitochondrial membrane, and it plays an important role in the membrane structure and function of mitochondria [27]. PC has multiple functions in anti-inflammation, cholesterol metabolism and neuronal differentiation in the CNS [28]. In PD, α-syn can promote the remodelling of pure phospholipid bilayers by weaker molecular interactions with PC and regulate cell membrane function [29]. In a rat 6-OHDA model, most PC species were found to be downregulated in the SN [30]. However, in the clinic, PD patients exhibited an increased PC intensity in serum compared with healthy controls [31]. These controversial results may be due to various differences between animal models and clinical trials [32]. Here, we focussed on CSF and found that it has increased expression of PC in PD. It has been shown that some PC species had increased levels in serum [31] but did not contain PC (O-18:3/20:3), PC (14:0/18:2) and PC (O-20:2/24:3) reported in CSF in this study. PC in serum may cross the BBB and contribute to its increased concentration in CSF, which makes the source of increased PC unclear [32]. To limit the influence of peripheral blood, this study excluded participants with metabolic diseases such as hyperlipidaemia. Nevertheless, more studies still need to confirm the relationship between lipids in CSF and the brain.

SM is a basic component of cellular membranes, including ceramide, ceramide-1-phosphate and sphingosine-1-phosphate, which are involved in inflammatory responses, cell death and autophagy [33]. In the CNS, SM can be hydrolysed into ceramide and phosphocholine, which are essential components of myelin in neurons [34]. SM is present in the myelin sheath and plays a role in presynaptic plasticity, nerve impulse transmission and neurotransmitter receptor localisation in the CNS [34]. The sphingomyelinase-1 dysfunction that induces lysosomal SM accumulation is a genetic risk factor for PD [35]. A recent study found that SM had no particular affinity for α-syn, but it can be found in α-syn-coated LBs [36]. In vitro, SM can increase the expression of α-syn in neurons [37]. In a clinical study, a special lipid profile with a significantly increased proportion of SM was found in fibroblasts from PD patients with the L444P GBA mutation [38]. In male PD patients, the level of sphingomyelin was significantly higher in the SN than in healthy controls, which was suggested to be attributed to its enrichment in LBs and α-syn production [39]. The characteristics of SM in serum and CSF in PD are still unclear. This study showed for the first time that PD patients had increased expression of SM in CSF, including SM (d18:1/20:1), SM (d18:2/14:0), SM (d18:0/16:0) and SM (d18:2/24:1). In another study, SM (d18:1/20:1) was found to have increased expression in serum from PD patients but not the whole SM family [40]. It was supposed that SM in CSF does not have a strong enough influence to change its concentration in serum. 

The CNS is the most cholesterol-rich organ, accounting for nearly 25% of the total amount [41]. Almost 80% of cholesterol in the brain is located in myelin [41]. CE is one form of cholesterol in the blood plasma and is a constituent of circulating HDL, LDL, VLDL and CM [42]. In the brain, CE is the storage form of cholesterol with levels less than 1% of TC, and it helps to transport cholesterol in the brain [42,43]. The delivery of cholesterol to neurons contributes to axonal regeneration, neurite extension and synaptogenesis in the brain [42]. Considering that cholesterol cannot cross the BBB under normal conditions, cholesterol maintenance in the brain mainly depends on glial cell in situ biosynthesis [24]. Thus, the impaired storage and delivery function of CE has a close relationship with neurological diseases [44]. In late-onset AD in patients and animal models, massive CE were found in the lesion region of the brain, especially in glial cells [45]. In AD patient-derived neurons, the increased CE is associated with tau pathology [45,46]. The triggering receptor expressed on myeloid cells 2 (TREM2), as a genetic susceptibility factor of AD, is required for cholesterol synthesis [47]. In TREM2 KO and AD-variant human iPSC microglia, impaired cells showed CE aggregates [47,48]. The cholesterol in intracellular membranes binds to the peptide domain of α-syn, and the cholesterol-rich regions may act as aggregation sites for α-syn in PD [49]. In the clinic, a higher concentration of cholesterol in serum was reported to be related to the slower clinical progression of PD [50]. Another study on PD patients with the GBA mutation found a lower level of cholesterol in serum compared with healthy controls [14,49]. The controversial result suggested that cholesterol metabolism could be regulated by different PD pathogeneses [51]. Regarding CE as a storage form of cholesterol, its profile and function in serum or CSF in PD still need further research. Here, we showed that PD patients had increased expression of CE in CSF, including CE (22:6) and CE (20:4). This finding supported the opinion of cholesterol disorder in the brains of PD patients and hints that CE in CSF may be a considerable biomarker for PD diagnosis. We analysed the relationship between TC in serum and CE in CSF. No association was found in healthy controls, which may be caused by the blocking of the healthy BBB, but a positive correlation was found in PD patients. This positive correlation could be explained by the BBB leakage associated with PD pathogenesis reported previously [52].

This study had some limitations. First of all, the study recruited a small number of subjects, including 17 with sporadic PD and 10 controls. More large-scale clinical trials still need to verify the robustness of the results. The subgroup analysis also faced the problem of small sample size. Secondly, lack of comparisons with other neurological diseases and small participant number in study impaired the clinical value of this lipid profile as a potential biomarker for PD diagnosis. Finally, an independent cohort design can provide more valuable results than the cross-sectional design here.

## 6. Conclusions

PD pathogenesis induces changes in the lipidomic profile in CSF, which has potential value as a diagnostic tool. This tool may help to more easily and precisely diagnose PD and effectively make therapeutic decisions. The PC, SM and CE in CSF that are altered at disease onset may open a new window for the evaluation of lipid disorders in the brains of PD patients and provide a new route into anti-PD drug discovery.

## Figures and Tables

**Figure 1 brainsci-13-01166-f001:**
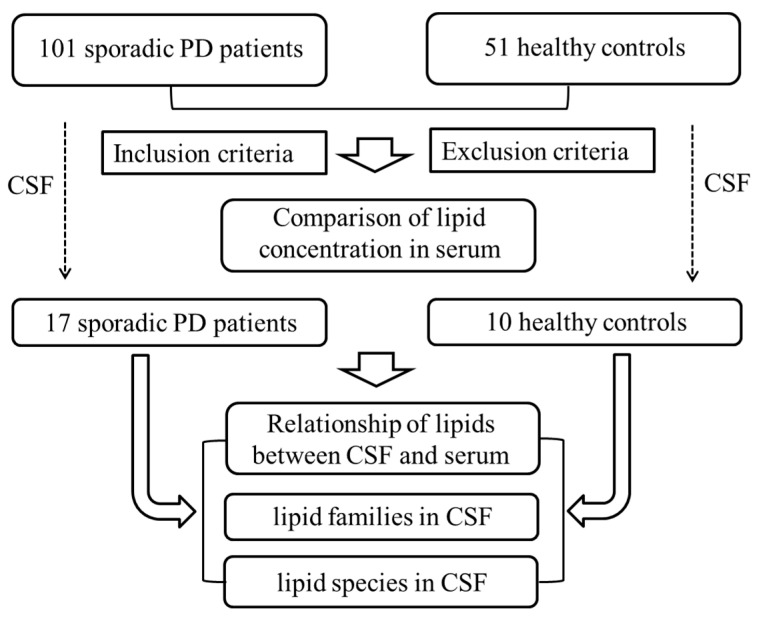
Flow chart of lipids identification strategy. Abbreviations: PD, Parkinson’s disease; CSF, cerebrospinal fluid.

**Figure 2 brainsci-13-01166-f002:**
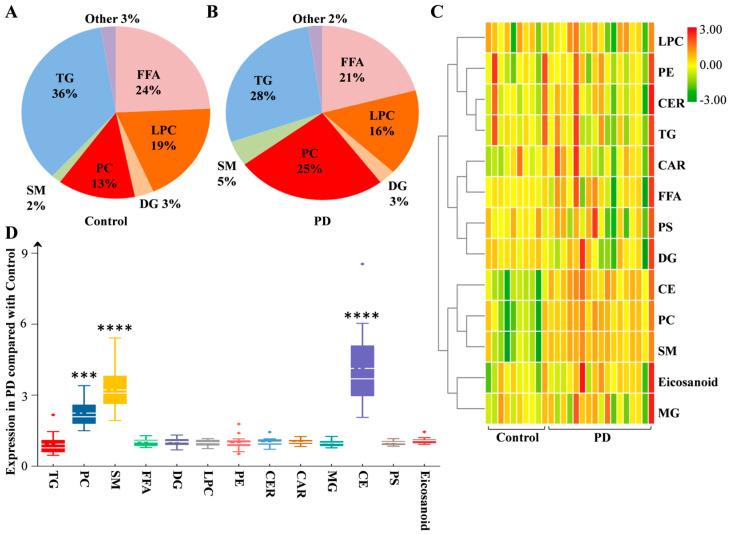
Lipid families from CSF in PD. The detected lipid species were classified into 13 families. According to the expression, the proportions of each lipid family in the control (**A**) and PD (**B**) groups are listed. (**C**) Heatmap representing all 13 lipid families obtained by UPLC–MS/MS. (**D**) The fold changes in the expression of lipid families between the PD and control groups are shown as box plots. Differences were determined by Student’s *t* test, with *** *p* < 0.001, **** *p* < 0.0001 by the Mann–Whitney test. The solid line in the box represents the mean lipid expression, and the dotted horizontal line represents the median lipid expression. Abbreviations: PD, Parkinson’s disease; TG, triglyceride; PC, phosphatidylcholine; SM, sphingomyelin; FFA, free fatty acid; DG, diglyceride; LPC, lysophosphatidylcholine; PE, phosphatidylethanolamine; CER, ceramide; CAR, 3-hydroxy-dodecanoyl-carnitine; MG, monoglyceride; CE, cholesterol ester; PS, phosphatidylserine.

**Figure 3 brainsci-13-01166-f003:**
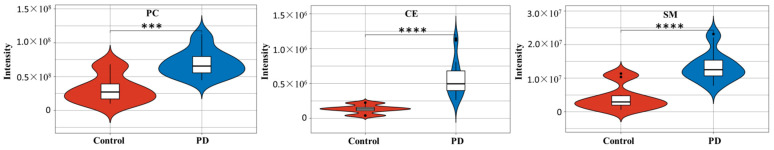
Characteristic lipid families from CSF in PD. Violin plot showing the expression and distribution of lipid families in the PD and control groups. Differences were determined by Student’s *t* test. Mean ± SD are shown, with *** *p* < 0.001, **** *p* < 0.0001 by the Mann–Whitney test. Abbreviations: PD, Parkinson’s disease; PC, phosphatidylcholine; SM, sphingomyelin; CE, cholesterol ester.

**Figure 4 brainsci-13-01166-f004:**
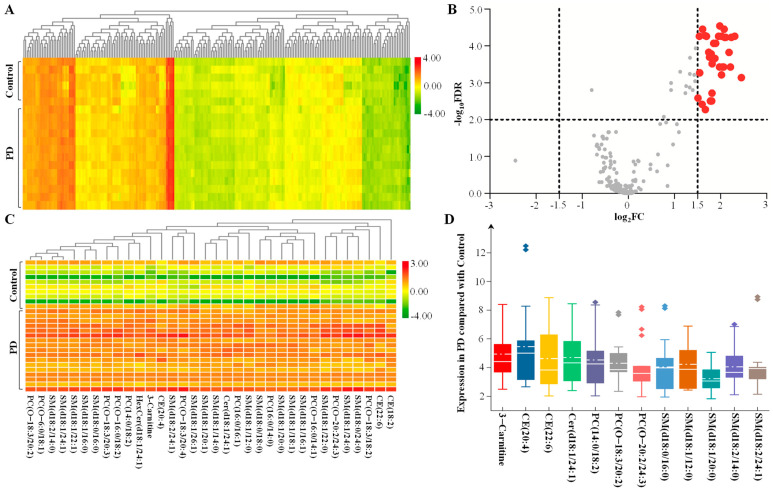
Lipid signature from CSF in PD. (**A**) Heatmap representing all 231 lipid species obtained by UPLC–MS/MS. (**B**) The volcano plot of the differential lipid species in the PD group compared with the control group. The red spots represent upregulated lipids that met −log_10_(FDR) > 2 and log_2_FC > 1.5, and grey dots represent insignificant differences. (**C**) The lipids with −log_10_(FDR) > 2 and log_2_FC > 1.5 were used to draw the characteristic lipidomic signature in the heatmap. (**D**) The fold change in the expression of lipids with log_2_FC > 2 is shown as a box plot. The solid line in the box represents the mean lipid expression, and the dotted horizontal line represents the median lipid expression. Abbreviations: PD, Parkinson’s disease; FC, fold change; FDR, false discovery rate; PC, phosphatidylcholine; SM, sphingomyelin; CER, ceramide; CE, cholesterol ester.

**Figure 5 brainsci-13-01166-f005:**
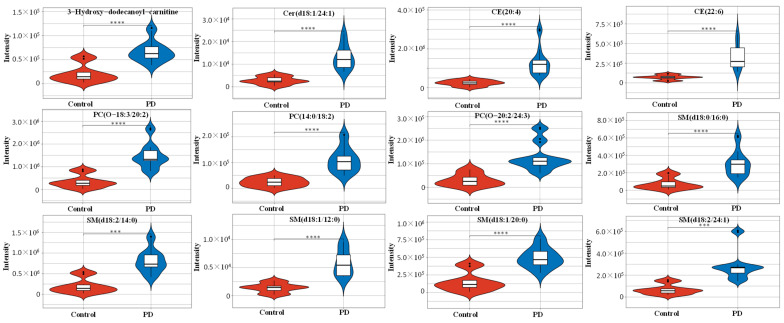
Characteristic lipid species from CSF in PD. Violin plot showing the expression and distribution of lipid species with −log_10_(FDR) > 2 and log_2_FC > 2. Differences were determined by Student’s *t* test. Means ± SDs are shown, with *** *p* < 0.001, **** *p* < 0.0001 by the Mann–Whitney test. Abbreviations: PD, Parkinson’s disease; PC, phosphatidylcholine; SM, sphingomyelin; CER, ceramide; CE, cholesterol ester.

**Figure 6 brainsci-13-01166-f006:**
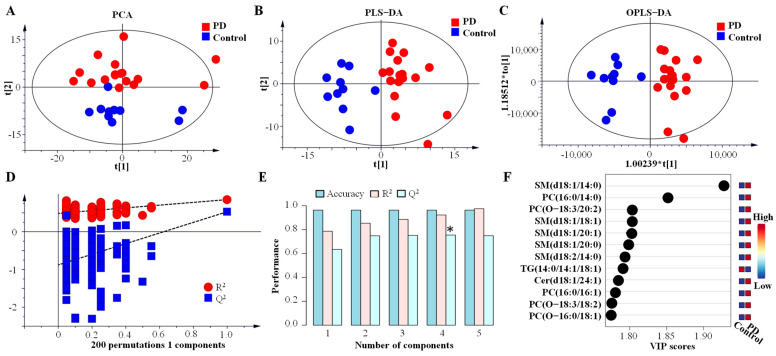
Multivariate statistics on lipidomic signature from CSF in PD. (**A**) Nonsupervised analysis by PCA. (**B**) Supervised analysis by PLS–DA. (**C**) Supervised analysis by OPLS–DA. (**D**) The permutation test of the OPLS–DA model. (**E**) Cross validation values of PLS–DA model, * *p* < 0.05 represent significant differences. (**F**) VIP score plot and identified lipid species with the top twelve values. Abbreviations: PD, Parkinson’s disease; PCA, principal component analysis; PLS–DA, orthogonal partial least-squares discriminant analysis; OPLS–DA, orthogonal partial least squares discriminate analysis; VIP, variable importance in projection; PC, phosphatidylcholine; SM, sphingomyelin; TG, triglyceride; CER, ceramide.

**Figure 7 brainsci-13-01166-f007:**
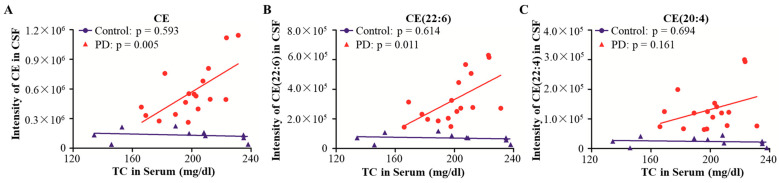
Correlation analysis between CE in CSF and TC in peripheral blood. The correlations between CE (**A**), CE (22:6) (**B**) and CE (20:4) (**C**) in CSF and TC in peripheral blood were analysed in the PD and control groups by Pearson correlation analysis. Abbreviations: PD, Parkinson’s disease; CE, cholesterol ester; TC, total cholesterol.

**Table 1 brainsci-13-01166-t001:** Lipidomic signature from serum in PD.

Demographics	Control (n = 51)	PD (n = 101)	*p*-Value
Gender (Female/Male)	20/31	39/62	0.540 ^a^
Age, years, mean (SD)	65.28 ± 8.78	66.65 ± 8.86	0.413 ^b^
Disease duration, years, mean (SD)	-	3.0 (0.5, 10)	-
Education, Junior high school or less, (%)	34 (66.67%)	73 (72.28%)	0.573 ^a^
UPDRS total score, media (min, max)	-	56 (16, 166)	-
H-Y stage, media (min, max)	-	2 (1, 5)	-
MMSE score, mean (SD)	27.78 ± 1.82	27.31 ± 1.93	0.147 ^b^
MOCA score, mean (SD)	28.41 ± 1.47	27.71 ± 1.46	0.006 ^b^**
TG (mg/dL), mean (SD)	109.20 ± 39.41	106.51 ± 33.51	0.663 ^c^
TC (mg/dL), mean (SD)	205.06 ± 31.78	204.46 ± 26.29	0.903 ^c^
HDL (mg/dL), mean (SD)	57.15 ± 10.83	58.32 ± 10.03	0.512 ^c^
LDL (mg/dL), mean (SD)	124.55 ± 28.52	122.66 ± 23.36	0.666 ^c^

^a^ Chi-square test. ^b^ Unpaired *t*-test. ^c^ Mann-Whitney test. ** *p* < 0.01 versus control. Abbreviations: PD, Parkinson’s Disease; SD, standard deviation; H&Y stage, Hoehn–Yahr stage; UPDRS, Unified Parkinson’s Disease Rating Scale; MMS, mini-mental state examination; MOCA, Montreal cognitive assessment; TG, triglyceride; TC, total cholesterol; HDL, high-density lipoprotein; LDL, low-density lipoprotein cholesterol.

**Table 2 brainsci-13-01166-t002:** Demographic characterisation of PD cases and controls for LC/MS.

Items	Control (n = 10)	PD (n = 17)	*p*-Value
Gender (Female/Male)	4/6	10/7	0.344 ^a^
Age (years)	63.50 ± 3.96	59.69 ± 6.70	0.226 ^b^
Disease duration (yeas)	-	2.0 (0.5, 4)	-
Education (Junior high school or less)	4 (66.67%)	11 (84.62%)	0.557 ^a^
UPDRS total score	-	50 (25, 87)	-
H-Y stage	-	2 (1, 2.5)	-
MMSE score	28.17 ± 1.34	27.31 ± 1.73	0.322 ^b^
MOCA score	28.83 ± 1.21	27.62 ± 1.33	0.089 ^b^
TG (mg/dL)	111.30 ± 44.29	97.95 ± 33.72	0.538 ^c^
TC (mg/dL)	197.61 ± 27.06	197.07 ± 18.97	0.963 ^c^
HDL (mg/dL)	54.41 ± 8.54	60.72 ± 3.87	0.404 ^c^
LDL (mg/dL)	128.51 ± 8.74	126.33 ± 15.67	0.767 ^c^

^a^ Chi-square test. ^b^ Unpaired *t*-test. ^c^ Mann-Whitney test. Abbreviations: PD, Parkinson’s Disease; SD, standard deviation; H&Y stage, Hoehn–Yahr stage; UPDRS, Unified Parkinson’s Disease Rating Scale; MMS, mini-mental state examination; MOCA, Montreal cognitive assessment; TG, triglyceride; TC, total cholesterol; HDL, high-density lipoprotein; LDL, low-density lipoprotein cholesterol.

## Data Availability

All data generated and analysed in the current study were collected from distinct subjects and are available from the corresponding author upon reasonable request.

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
