# Peer review of "Lipid Metabolism Disorder in Cerebrospinal Fluid Related to Parkinson’s Disease"

_brainsci, 2023, doi:10.3390/brainsci13081166_

Round 1

Reviewer 1 Report

In the manuscript submitted by Qiu et. al. titled as ‘Lipid metabolism disorder in cerebrospinal fluid related to Parkinson’s disease’, the authors conducted lipid profiling in the serum and CSF samples from PD patients and healthy control. They observed an increase of CE, PC, SM lipids in CSF. The correlation between CSF-CE or CSF-CE (22:6) and total cholesterol in the serum of the PD samples indicated an impairment of BBB.

Overall, this paper is of strong novelty to fill in the knowledge gap which is the CSF-lipid profile in PD patients. The paper itself is well written and organized with clear logic. I only have two suggestions.

1. The authors stated that the differential abundant lipids in PD-CSF samples may be considered as biomarkers. Accordingly, it is suggested to perform correlation analysis  between each of all the lipids or signature peptides and PD symptom readouts.

2. Please use a different colormap instead of green-red, like red-blue, for the hierarchical clustering figures in Fig. 3C, 5A&C.

Author Response

Thanks for the comments to improve our work.

  • We agree that correlation analysis should been performed. It is a limitation that samples are not enough for correlation analysis in our study. Thus, we make a conclusion that "the lipidomic signature in CSF may be considered a potential biomarker ".

Re: We have replaced them following the suggestion.

  • Please use a different colormap instead of green-red, like red-blue, for the hierarchical clustering figures in Fig. 3C, 5A&C.

Re: We have redrawn the figure.

Reviewer 2 Report

The study aimed to analyze lipid metabolism in the cerebrospinal fluid (CSF) of Parkinson's disease (PD) patients and healthy controls without peripheral blood dyslipidemia. The study found that lipid families, such as phosphatidylcholine (PC), sphingomyelin (SM), and cholesterol ester (CE), had increased expression in PD patients compared to controls. The researchers utilized hierarchical clustering to differentiate PD patients based on the altered expression of 34 lipid species and identified 12 lipid species to establish the lipidomic signature of PD. The study also revealed a positive correlation between the CE family and CE in CSF and total cholesterol in the peripheral blood of PD patients. These findings suggest that the lipidomic signature in CSF could potentially serve as a biomarker for PD diagnosis, and increased CE, PC, and SM in CSF may indicate pathological changes in the brains of PD patients. The research is crucial in understanding the mechanism of Parkinson's disease.

In the "Lipid composition from CSF in PD" subsection, the authors mentioned Figures 1 A, B, C, and D, but I could not find these panels in Figure 1.

It is recommended to provide the exact p-value in Figures 4 and 5.

Please specify the R-value in Figure 7.

Regarding Figure 5b, could you comment on the separability of the two clusters of the PD group and the control group?

Please format the references appropriately.

Author Response

Thanks for the comments to improve our work.

  • In the "Lipid composition from CSF in PD" subsection, the authors mentioned Figures 1 A, B, C, and D, but I could not find these panels in Figure 1.

Re: The “Figure 1” has been replaced with “Figure 2” to correct the mistake.

  • It is recommended to provide the exact p-value in Figures 4 and 5.

Re: We have supplied the p-value in result part in Page 5.

  • Please specify the R-value in Figure 7.

Re: We have supplied the p-value in result part in Page 5.

  • Regarding Figure 5b, could you comment on the separability of the two clusters of the PD group and the control group?

Re: Figure 5 did not contain “b”. Is it Figure 6b? We have explained as “In supervised methods, such as PLS-DA, 20.1% of the total variance by component-1 and 19.7% of the total variance by component-2 can be explained (Figure 6 B). It showed a valuable clusterization of PD and healthy subjects with a specific lipidomic signature” in “Different pattern recognition for the lipidomic signature in CSF” section in Page.5.

  • Please format the references appropriately.

Re: We have formated the references.

Reviewer 3 Report

Accept in present form

Accept in present form

Author Response

Thanks for your support.

Reviewer 4 Report

The authors have investigated the lipid profile from CSF of Parkinson’s disease patients and controls.  In general, the concept is reasonable (except that they have not correlated the CSF results to serum lipids with the exception of CE in CSF to TC in serum) and the need for improved biomarkers is clear.  However, this study has a number of significant limitations which at the very least should be acknowledged and discussed, but ideally additional samples should be collected and analyzed.

1)      The authors refer to this profile as a potential biomarker for PD diagnosis.  They cannot do this without evaluating a similar (at minimum) number of samples from other neurological conditions to demonstrate that this profile is indeed selective / enriched in PD patients.

2)      The study is cross-sectional and has a small number of subjects (17 sporadic PD and 10 controls).  Increasing the ‘n’ and collecting samples from patients later in disease would provide important insights into whether this profile may reflect disease progression. 

3)      Confirmation in an independent cohort will be required.

4)      The authors state that CSF represents the brain.  This should be demonstrated by analyzing post-mortem brain samples and comparing / correlating the profiles to both ante-mortem and post-mortem CSF samples.

Specific / technical points:

·         It is not clear that how the data were acquired, unbiased or targeted?

·         It would be clearer if the authors can describe how those lipids’ structures were assigned, confirmed, normalized and quantified,  3-hydroxy-dodecanoyl-carnitine, Cer (d18:1/24:1), CE(20:4), CE(22:6), PC(14:0/18:2), PC(O-18:3/20:2), PC(O-20:2/24:3), SM(d18:0/16:0), SM(d18:2/14:0), SM(d18:2/24:1), SM(d18:1/20:1), and SM(d18:1/12:0).

·         Figure 7 shows a lack of correlation between CSF CE and serum TC for controls, yet a strong correlation for PD.  What is the correlation between CSF CE and serum CE?  Related point: Lipid families: CE, PC and SM showed statistically significant increases (> 2x fold) in PD.  Would those observation also translated into plasma? Have the authors analyzed the plasma/serum samples corresponding to these 27 subjects?   These data do not appear to be included and should be added.

·         In scheme 3, please add a column for m/z of lipid species.  

·         More details are required in the methods section on the CSF collection, processing, QC and storage.

Page 2: "More seriously, research on lipid metabolism" - re-word, inappropriate use of 'more seriously'.

Page 6: "Lipids, related to vesicles and membrane fragments, are found within α-syn-coated LBs in the postmortem PD brain and may be released into CSF" - this is not accurate, there is no direct measurement of this process.  This should be reworded.

Page 6: "In a rat model insulted by 6-hydroxydopamine," can simply be stated as 'In a rat 6-OHDA model......'

Page 6: " More details of SM functions and expression in PD still need further study" should be re-worded.

Author Response

Thanks for the comments to improve our work.

  • The authors refer to this profile as a potential biomarker for PD diagnosis.  They cannot do this without evaluating a similar (at minimum) number of samples from other neurological conditions to demonstrate that this profile is indeed selective / enriched in PD patients.

Re: We agree with the suggestion that other neurological disease should be used to compare. Considered the limitation of this study, we only suggested that "the lipidomic signature in CSF may be considered a potential biomarker" in conclusion to avoid the confusion.

  • The study is cross-sectional and has a small number of subjects (17 sporadic PD and 10 controls).  Increasing the ‘n’ and collecting samples from patients later in disease would provide important insights into whether this profile may reflect disease progression. 

Re: Good suggestion. We will collect more samples in future study.

  • Confirmation in an independent cohort will be required.

Re: we have added it as “A total of 152 Han Chinese subjects, including 101 sporadic PD patients and 51 healthy controls matched by age and sex, were recruited from the First Affiliated Hospital of Guangzhou Medical University from January 2018 to August 2021 as an independent cohort” in the method part.

  • The authors state that CSF represents the brain.  This should be demonstrated by analyzing post-mortem brain samples and comparing / correlating the profiles to both ante-mortem and post-mortem CSF samples.

Re: we have replaced with “Our results revealed that the lipidomic signature in CSF may be considered a potential biomarker for PD diagnosis, and increased CE, PC and SM in CSF may reveal pathological changes in PD patients, such as blood‒brain barrier leakage.”

  • Specific / technical points: It is not clear that how the data were acquired, unbiased or targeted?

Re: we have supplied as “A widely targeted metabolome method based on the Metware database (Metware Biotechnology Co., Ltd, Wuhan, Hubei, China) was performed in UPLC‒MS/MS analysis” in “lipidomic analysis” in method part.

  • It would be clearer if the authors can describe how those lipids’ structures were assigned, confirmed, normalized and quantified, 3-hydroxy-dodecanoyl-carnitine, Cer (d18:1/24:1), CE(20:4), CE(22:6), PC(14:0/18:2), PC(O-18:3/20:2), PC(O-20:2/24:3), SM(d18:0/16:0), SM(d18:2/14:0), SM(d18:2/24:1), SM(d18:1/20:1), and SM(d18:1/12:0).

Re: we have supplied as “A widely targeted metabolome method based on the Metware database (Metware Biotechnology Co., Ltd, Wuhan, Hubei, China) was performed in UPLC‒MS/MS analysis” in “lipidomic analysis” in method part.

  • Figure 7 shows a lack of correlation between CSF CE and serum TC for controls, yet a strong correlation for PD. What is the correlation between CSF CE and serum CE?  Related point: Lipid families: CE, PC and SM showed statistically significant increases (> 2x fold) in PD. Would those observation also translated into plasma? Have the authors analyzed the plasma/serum samples corresponding to these 27 subjects? These data do not appear to be included and should be added.

Re: It is a good suggestion. We did not detect the lipidomic analysis in serum by UPLC‒MS/MS. We only detected the lipid in serum by a ChemWell chemistry analyser to exclude lipid disorder in serum. The correlation between CSF CE and serum CE were excluded without significant difference, and theis results were not shown in manuscript.

  • In scheme 3, please add a column for m/z of lipid species.

Re: we have added the m/z column following the suggestion.

  • More details are required in the methods section on the CSF collection, processing, QC and storage.

Re: we have added the details as following “The CSF sample was thawed on ice, whirl around 10 s, and then centrifuge it with 3000 rpm at 4 ℃ for 5 min. Take 50 uL of one sample and homogenize it with 1mL mixture (include methanol,MTBE and internal standard mixture ). Whirl the mixture for 2 min. Then add 500 uL of water and whirl the mixture for 1 min, and centrifuge it with 12,000 rpm at 4 ℃ for 10 min. Extract 500 uL supernatant and concentrate it. Dissolve powder with 100 uL mobile phase B ,then stored in -80 ℃. Finally take the dissolving solution into the sample bottle for LC-MS/MS analysis. The maximum number of freeze‒thaw cycles for detection was 2”.

  • Page 2: "More seriously, research on lipid metabolism" - re-word, inappropriate use of 'more seriously'.

Re: We have deleted it.

  • Page 6: "Lipids, related to vesicles and mem[1]brane fragments, are found within α-syn-coated LBs in the postmortem PD brain and may be released into CSF" - this is not accurate, there is no direct measurement of this process.  This should be reworded.

Re: We have deleted this sentence.

  • Page 6: "In a rat model insulted by 6-hydroxydopamine," can simply be stated as 'In a rat 6-OHDA model......'

Re: We have replaced it.

  • Page 6: " More details of SM functions and expression in PD still need further study" should be re-worded.

Re: We have deleted this sentence to avoid confusion.

Round 2

Reviewer 4 Report

I should have been more explicit in my first review.  The limitations that I pointed out* need to be acknowledged and discussed in the manuscript if they are not going to add key samples and perform additional experiments.  Simply deleting a couple of sentences is not sufficient to address these shortcomings that need to be attended in writing in the manuscript.

* However, this study has a number of significant limitations which at the very least should be acknowledged and discussed, but ideally additional samples should be collected and analyzed.

11)   The authors refer to this profile as a potential biomarker for PD diagnosis.  They cannot do this without evaluating a similar (at minimum) number of samples from other neurological conditions to demonstrate that this profile is indeed selective / enriched in PD patients.

22) The study is cross-sectional and has a small number of subjects (17 sporadic PD and 10 controls).  Increasing the ‘n’ and collecting samples from patients later in disease would provide important insights into whether this profile may reflect disease progression. 

   3) Confirmation in an independent cohort will be required.

4 4) The authors state that CSF represents the brain.  This should be demonstrated by analyzing post-mortem brain samples and comparing / correlating the profiles to both ante-mortem and post-mortem CSF samples.

The authors have completely missed the point in #3 re: confirmation in an independent cohort.  This means a second set of independent samples run through the same experimental conditions to confirm the original results.

The new text in the methods section needs addressing:

"“The CSF sample was thawed on ice, whirl around 10 s, and then centrifuge it with 3000 rpm at 4 for 5 min. Take 50 uL of one sample and homogenize it with 1mL mixture (include methanolMTBE and internal standard mixture ). Whirl the mixture for 2 min. Then add 500 uL of water and whirl the mixture for 1 min, and centrifuge it with 12,000 rpm at 4 for 10 min. Extract 500 uL supernatant and concentrate it. Dissolve powder with 100 uL mobile phase B ,then stored in -80 . Finally take the dissolving solution into the sample bottle for LC-MS/MS analysis. The maximum number of freeze‒thaw cycles for detection was 2”."

Author Response

11)   The authors refer to this profile as a potential biomarker for PD diagnosis.  They cannot do this without evaluating a similar (at minimum) number of samples from other neurological conditions to demonstrate that this profile is indeed selective / enriched in PD patients.

RE: we agree that small sample size cannot draw a precise profile as a potential biomarker for PD. The main opinion in our study is showed that lipid disorder in CSF may occur in PD patients, and this lipid disorder may reflect the pathological change in the brain, and may have potential diagnosis value in clinic. This is a support evidence for the fundamental study in future.

We have added the drawbacks of this study in the discussion part as following:

This study had some limitations. First of all, the study recruited a small number of subjects, included 17 sporadic PD and 10 controls. More large-scale clinical trials still need to verify the robustness of the results. The subgroup analysis also faced the problem of small sample size. Secondly, lack of comparisons with other neurological diseases impaired the clinical value of this lipid profile as a potential biomarker for PD diagnosis. Finally, an independent cohort design can provide more valuable results than the cross-sectional design here.

22) The study is cross-sectional and has a small number of subjects (17 sporadic PD and 10 controls).  Increasing the ‘n’ and collecting samples from patients later in disease would provide important insights into whether this profile may reflect disease progression. 

RE: We agree that more samples can provide more details. To avoid the confounding factor, we excluded the PD patients with metabolic diseases or lipid-related gene mutation such as GBA. This exclusion criterion makes the result more reliable, but limits the recruited samples.

   To avoid the batch-to-batch effects, the UPLC‒MS/MS analysis in this study was detected in the same batch [1]. Collecting more samples for the next analysis may increase the error of batch-to-batch effects.

According to the result of sample size estimation,the sample size is enough for this study, but not enough for the subgroup analysis.

[1] Fu X, Anderson M, Wang Y, et al. LC-MS/MS-MRM-based targeted metabolomics for quantitative analysis of polyunsaturated fatty acids and oxylipins[J]. High-Throughput Metabolomics: Methods and Protocols, 2019: 107-120.

3 3) Confirmation in an independent cohort will be required.

RE: We have corrected it with “cross-sectional study” to avoid the confusion.

4 4) the authors state that CSF represents the brain.  This should be demonstrated by analyzing post-mortem brain samples and comparing / correlating the profiles to both ante-mortem and post-mortem CSF samples.

RE: Thanks for the suggestion. We do not use post-mortem brain here. The major reason is that the blood-CSF-barrier (BBB) may have broken after brain death. Thus, the molecular in CSF from post-mortem brain may be affected by serum.

We suggest that CSF may represent the brain, based on the reasons as follow:

  • The Parkinson Study Group has reported that homovanillic acid (HVA) concentration in CSF almost doubled in PD group and possibly could reflect compensatory processes among surviving dopaminergic neurons of the PD brain in 1992. And now, CSF HVA concentration is considered ideal biomarker for PD diagnosis [1].
  • A serial studies showed a-syn [2], miRNAs [3], tau [4] in CSF may be used as biomarker of PD.
  • A serial studies showed that CSF lipidomics may act as biomarkers of Alzheimer's disease [5], ALS [6] and MS [7]
  • The breakdown in BBB can induce the different changed lipid profiles in CSF and serum. It also may take action in PD [8].

Thus, we suggested the lipids in CSF may be a potential biomarker of PD.

[1] LeWitt P A, Galloway M P, Matson W, et al. Markers of dopamine metabolism in Parkinson's disease[J]. Neurology, 1992, 42(11): 2111-2111.

[2] Oligomeric and phosphorylated alpha-synuclein as potential CSF biomarkers for Parkinson's disease[J].Molecular Neurodegeneration, 2016, 11.DOI:10.1186/s13024-016-0072-9.

[3] Tainá M,Marques, Bea H ,et al.MicroRNAs in Cerebrospinal Fluid as Potential Biomarkers for Parkinson's Disease and Multiple System Atrophy.[J].Molecular neurobiology, 2017.DOI:10.1007/s12035-016-0253-0.

[4] Shi M , Zhang J .CSF α-synuclein, tau, and amyloid β in Parkinson\"s disease[J].The Lancet Neurology, 2011, 10(8):681-.DOI:10.1016/s1474-4422(11)70130-2.

[5] Michelle,Mielke,Norman,et al.CSF sphingolipid levels correlate with CSF beta-amyloid and tau[J].Alzheimers & Dementia, 2013.DOI:10.1016/j.jalz.2013.05.361.

[6] Blasco, H., Veyrat-Durebex, C., Bocca, C. et al. Lipidomics Reveals Cerebrospinal-Fluid Signatures of ALS. Sci Rep 7, 17652 (2017). https://doi.org/10.1038/s41598-017-17389-9.

[7] Pieragostino, D., Cicalini, I., Lanuti, P. et al. Enhanced release of acid sphingomyelinase-enriched exosomes generates a lipidomics signature in CSF of Multiple Sclerosis patients. Sci Rep 8, 3071 (2018). https://doi.org/10.1038/s41598-018-21497-5

[8] Padilla-Docal, B., Dorta-Contreras, A.J., Bu-Coifiu-Fanego, R. et al. CSF/serum quotient graphs for the evaluation of intrathecal C4synthesis. Fluids Barriers CNS 6, 8 (2009). https://doi.org/10.1186/1743-8454-6-8
